# eHealth Platforms to Promote Autonomous Life and Active Aging: A Scoping Review

**DOI:** 10.3390/ijerph192315940

**Published:** 2022-11-29

**Authors:** Joana Bernardo, João Apóstolo, Ricardo Loureiro, Elaine Santana, Nilufer Korkmaz Yaylagul, Carina Dantas, Filipa Ventura, Filipa Margarida Duque, Nina Jøranson, Minna Zechner, Willeke van Staalduinen, Vincenzo De Luca, Maddalena Illario, Rosa Silva

**Affiliations:** 1Health Sciences Research Unit: Nursing (UICISA: E), Nursing School of Coimbra (ESEnfC), 3030 Coimbra, Portugal; 2Portugal Centre for Evidence Based Practice: A JBI Centre of Excellence (PCEBP), 3030 Coimbra, Portugal; 3Gerontology Department, Faculty of Health Science, Akdeniz University, Antalya 07070, Turkey; 4SHINE 2Europe, 3030 Coimbra, Portugal; 5Faculty of Health Studies, VID Specialized University, N-3019 Oslo, Norway; 6Faculty of Social Sciences, University of Helsinki, 00014 Helsinki, Finland; 7AFEdemy—Academy on Age-Friendly Environments in Europe, 2806 ED Gouda, The Netherlands; 8Dipartimento di Sanità Pubblica, Università degli Studi di Napoli Federico II, 80138 Naples, Italy

**Keywords:** eHealth strategies, personal autonomy, healthy aging, SHAFE

## Abstract

New technologies, namely eHealth platforms, are being used more than ever before. These platforms enable older people to have a more independent lifestyle, enhance their participation, and improve their well-being. Information and communication technologies are expected to be linked to the triad of aging, social inclusion, and active participation, which is in line with the implementation of Smart Healthy and Age-Friendly Environments. This scoping review aimed to map eHealth platforms designed to promote autonomous life and active aging. The Joanna Briggs Institute methodology and the PRISMA-ScR checklist were used. A search was conducted on MEDLINE (via PubMed), CINAHL Complete (via EBSCOhost), Scopus, Cochrane Database of Systematic Reviews (via EBSCOhost), SciELO, DART-Europe, CAPES, and MedNar databases. Fourteen studies were included. This scoping review synthesized information on eHealth platforms designed to promote active living, their domains of intervention, and the outcomes assessed in those studies that have implemented and evaluated these eHealth platforms.

## 1. Introduction

It is widely recognized that health systems must shift from reactive disease management to health promotion and disease prevention, from a focus on illness to a focus on well-being, and from fragmentation to the integration of services along the continuum of care. These issues are not recognized in the European Union when only 3% of the health budgets focus on promoting health, independence, and autonomy [1].

Health promotion and disease prevention programs implemented by health professionals have the potential to reduce the risk progression from lower (i.e., no/few risks or disease absence) to higher risk status (i.e., added health risks and/or newly diagnosed disease) of individuals throughout the life cycle. Improved health over time will provide quality life years, reduce the prevalence of chronic disease and disability, as well as mitigate health care expenditures [2].

The concept of healthy and active aging is multidimensional and comprises bio-psycho-social-spiritual health. In a salutogenic perspective, aging is an achievement to be embraced, recognizing aging as a positive developmental process of a person. Accordingly, the life experience of the person is valued and older people are seen as assets for their wealth of experiences, resources, skills and knowledge. The salutogenic perspective involves a conscious reflection on how the individual considers life as a whole in order to cope with the inherent vulnerabilities and stressors of old age. However, the research endeavors to operationalize the salutogenic theory in interventions for older adults to age in an healthy manner is scarce [3].

The United Nations [4]’ agenda for sustainable development comprises 17 Sustainable Development Goals (SDGs) to be reached until 2030. More specifically, goal 3, relating to good health and well-being, and goal 11 on making cities and communities inclusive, safe, sustainable, and age-friendly by ensuring a better future for the population, will contribute to leveraging the relevance of active aging in today’s societies. 

Active aging is applicable both at the individual- and population level, aiming at allowing people to accomplish their full potential in physical, social, and mental well-being throughout life [5], and enabling their participation in society according to their needs, desires, and capabilities [6]. According to the World Health Organization (WHO) [7], the concept of active aging must be universal to promote the implementation of environments that allow individuals to overcome the challenges associated with this stage of life. The implementation of active aging allows overcoming the challenges of the increasingly aging population, while promoting a positive aging for individuals, communities, and societies. Accordingly, a set of interventions to optimize opportunities for health, participation, and security are delivered based on the promotion of healthy lifestyles, physical activity, and social interaction [8,9]. Thus, promoting and adopting healthier and participative lifestyles is essential to prevent disability and promote well-being. Initiatives that foster healthy environments where such changes may take place are needed.

Over the last decade, the rapid development of information and communication technologies (ICTs) in health has brought about a significant revolution in the way health and health services address the challenges of providing high-quality, effective, and safe care [10,11,12]. This technology-led transition is an adjustment in the way health information is captured, viewed, processed, and stored. This transition plays a key role in improving the quality of services, increasing coordination between providers, patients, and caregivers, improving patient management, helping to overcome physical distances, and involving patients in their own health and well-being [13].

Digital technology, namely ICTs, plays a facilitating role in society, as it can contribute to the well-being of older adults and enable them to age with dignity [14,15]. In other words, digital technology can facilitate health promotion, and has the potential to support individual and community engagement, and underpin healthy and active aging [16,17].

Therefore, it is imperative to adopt ICTs in the health field. In the era of global aging, digital technology is seen as a new opportunity to overcome several challenges associated with aging and promote older people’s healthy lifestyles and social inclusion [12,15,18]. According to the latest Digital Economy Outlook report of the Organization for Economic Co-operation and Development (OECD), 78.3% of adults aged 50–74 years are now connected to the Internet [19].

The COVID-19 pandemic has further highlighted the need for therapeutic interventions through digital platforms. Such platforms allow access to various health services that might otherwise be inaccessible. Additionally, they were an alternative to the de-livery of group-interventions, which had been severely hampered due to social restrictions [20,21,22,23,24,25].

The WHO [26] defines eHealth as the use of ICTs in support of health. eHealth platforms have been successful in promoting adherence to active and healthy lifestyles and improving self-management, particularly among older people [11,22,23,24,25,26].

During the early 2000s, the Internet was integrated into the everyday life of the general public. At the same time, eHealth platforms emerged as a new concept, expanding the focus of medical informatics to include clinical information systems [15].

eHealth platforms should focus on meeting the needs of an aging population by supporting independent living in place, self-management of age-related conditions, reducing isolation, and promoting active and healthy lifestyles [11,15,24,25].

Active and healthy living is a concept applicable to both individuals and population groups. Initiatives are needed to create healthy environments that promote healthy and participatory lifestyles, which essential for preventing disability and increasing well-being [26]. Therefore, more knowledge is essential to understand the relevance of the active aging process and the role of eHealth platforms will play in contributing to social inclusion and promoting individual health and well-being.

These considerations were shared by the European Innovation Partnership on Active and Healthy Aging (EIP-AHA), an initiative that aims to foster the innovative use of digital for active and healthy aging [27,28]. In the aftermath of the EIP-AHA, other initiatives continued the approach, evolving to a concept of Active and Healthy Living, where prevention starts from birth.

Moreover, patient engagement is a broader concept that encompasses adherence, empowerment, health literacy, shared decision-making, and patient activation. Shared decision-making tools can support patients in making decisions about their health, and eHealth platforms can support patients in communicating with peers and healthcare professionals, in addition to promoting self-management and health literacy [1,10].

The importance of engaging older adults in their own care is gaining increasing attention from health professionals, researchers, and policymakers, as their involve-ment towards health promotion and disease prevention is crucial to successful health management. An informed and active older adult is an essential partner in care. The engagement behavior framework suggests that older adults benefit from being en-gaged, and current healthcare delivery demands from them the skills to participate constructively in planning their health plan [29].

Some studies [16,30,31,32] report that older people who live alone and have limited engagement in work and social activities felt more socially connected, improved their well-being, and felt less lonely and socially isolated after using digital technologies. Other studies report use of ICTs among older adults is associated with preventive health behaviors and improved cognitive function [16,32]. Without neglecting the importance of social interaction as a human need across age groups, digitalization may become a powerful asset to support people in meeting their needs and foster well-being, ultimately creating more equitable and participatory societies [24,25].

In this approach, ICTs are expected to be linked to the triad of aging, social inclusion, and active participation, which is aligned with the implementation of Smart Healthy Age-Friendly Environments (SHAFE) [33]. The focus should be on people, with citizenship and social interaction as key elements, and places, such as houses, built environments, or community spaces. Thus, SHAFE brings many benefits to the well-being, participation, and active aging of all citizens [33].

The SHAFE network aims to facilitate the creation of healthy and friendly environments for all ages through the use of new technologies, emphasizing the importance of people and Places in creating better digital health solutions accessible to all. Placing the person at the center of the digitization process is crucial for implementing the strategy advocated by SHAFE [33], particularly regarding the use of eHealth platforms.

The literature consulted on the promotion of active aging revealed several studies on the planning and implementation of eHealth platforms. However, these studies vary according to their purpose, target population, platform use, areas of intervention, and outcomes assessed. Thus, it is urgent to map the information on eHealth platforms. In a preliminary search of MEDLINE (via PubMed), the Cochrane Database of Systematic Reviews, the JBI Evidence Synthesis, the Open Science Framework, and Prospero, no ongoing scoping reviews or scoping review protocols were identified, reinforcing the need to map the existing evidence on eHealth platforms to promote active aging. Therefore, the following research questions were explored:-Which eHealth platforms facilitate autonomous life and promote active aging?-What are the age groups targeted by the eHealth platforms?-What are the domains of intervention of these eHealth platforms (physical, emotional, cognitive, or social)?-Which outcomes are assessed in studies that have implemented and evaluated these eHealth platforms?

## 2. Materials and Methods

This scoping review follows the JBI methodology for scoping reviews (Peters et al., 2020) and meets the Preferred Reporting Items for Systematic Reviews and Meta-Analysis extension for Scoping Reviews (PRISMA-ScR) checklist [34]. The protocol of this review has already been published [35].

### 2.1. Search Strategy

A three-phase approach was implemented to locate both published and unpublished studies. An initial limited search of MEDLINE (via PubMed) and CINAHL (via EBSCOhost) databases was undertaken to identify the text words in the titles and abstracts of relevant articles (first phase). The keywords/text words in the titles and abstracts of relevant articles and the index terms used to describe the articles were used to develop a full search strategy (second phase). The search strategy, including all identified keywords and index terms, was adapted to each database. The reference lists of all included studies were screened for additional studies (third phase).

The databases used for search included MEDLINE (via PubMed), CINAHL Complete (via EBSCOhost), Scopus, Cochrane Database of Systematic Reviews (via EBSCOhost), and SciELO. Sources of unpublished studies used for searches included DART-Europe, CAPES, and MedNar.

The search strategy was limited to studies published in Portuguese, Spanish, and English, without time constraints.

Following the JBI methodology for scoping reviews, the P (participants), C (concept), and C (context) mnemonic was applied to define the following inclusion criteria: Participants: studies involving people, regardless of age; Concept: studies on eHealth platforms, designed and/or created to promote autonomous life and/or active aging; Context: studies conducted in all contexts. As for the exclusion criteria, this scoping review did not consider studies addressing eHealth platforms exclusively for educating and assisting health professionals in decision-making, chronic or acute disease surveillance, and monitoring, and access to person-directed medical information. Quantitative, qualitative, or mixed-methods studies were considered for inclusion in this review.

The search strategy is described in Appendix A.

### 2.2. Study Selection and Data Extraction

All studies identified through database searching were retrieved and stored in Mendeley^®^ V1.19.8 (Mendeley Ltd., Elsevier, The Netherlands) and duplicates were removed. Then, the articles were imported into Rayyan QCRI (Qatar Computing Research Institute [Data Analytics], Doha, Qatar). Two independent reviewers screened the titles and abstracts of the identified studies to determine if they met the inclusion criteria.

The full texts of eligible studies were retrieved. Two independent reviewers assessed them in detail against the inclusion criteria. Studies that did not meet the inclusion criteria were excluded. Finally, the reference lists of all included studies were screened.

Any disagreements that arose between the reviewers were resolved through discussion or with the intervention of a third reviewer at each stage of the study selection process. The original authors were contacted if any full-text version was unavailable or in case of missing data. The search results are reported in full in Table 1 [34].

Finally, two independent reviewers extracted data from the studies using the methodology proposed by the JBI [36] and based on the review objectives and questions.

## 3. Results

### 3.1. Study Characteristics, Settings, and Samples

Of the 518 potentially relevant studies (after removing duplicates), 483 studies were excluded after title and abstract screening. The full-text versions of the remaining 35 articles were read and 14 were found to meet the inclusion criteria (10 resulting from database searching and 5 from the analysis of secondary sources) (Figure 1).

The included 14 articles were published between 2008 and 2021. They were conducted in several countries: Consortium Italy, Switzerland, Spain, and Romania (*n* = 1) [37]; The Netherlands (*n* = 1) [38]; USA (*n* = 1) [39]; Consortium Spain, USA, Greece, Ireland, and Italy (*n* = 1) [40]; Consortium UK, The Netherlands, Spain, Germany, Austria, France, and Belgium (*n* = 1) [41]; Greece (*n* = 1) [42]; USA (*n* = 1) [43]; Germany (*n* = 1) [44]; Consortium Spain, Sweden, and Greece (*n* = 1) [45]; Belgium (*n* = 1) [46]; Consortium The Netherlands and Italy (*n* = 1) [47]; Consortium Switzerland and The Netherlands (*n* = 1) [48]; Consortium Norway, Germany, and The Netherlands (*n* = 1) [49]; Italy (*n* = 1) [50]. Five studies were experimental studies (Randomized Controlled Trials—RCTs) [38,43,44,45,49], eight studies were descriptive studies [37,39,40,41,42,46,47,50], and one was a cross-sectional study [48] Table 1.

The following age ranges were found in these studies: over 40 years [38], over 55 years [43], over 60 years [46,49], and over 65 years [37,39,42,44,45,50]. One only mentioned the group of older adults, without specifying an age limit [40,41,47,48].

**Table 1 ijerph-19-15940-t001:** Research articles included in this scoping review.

Author, Year, Country	Project/Platform Development	Aims/Purpose	Age Group (Years)/Sample Size	eHealth Platforms/Project
Palumbo et al. (2020) [37]. Consortium Italy, Switzerland, Spain, Romania	This paper describes the design of the NESTORE methodology and its IoT architecture.	To describe the design of the NESTORE methodology and its IoT architecture and describe the validation strategy to assess the effectiveness of NESTORE as a coaching platform for healthy aging.	65–75/*	NESTORE system—coaching activities and personalized feedback to the user.
Reijnders et al. (2015) [38]. The Netherlands	A randomized controlled trial	To investigate the effectiveness of the eHealth intervention in terms of subjective cognitive functioning and to measure objective cognitive functioning and psychological well-being.	40–65/*n* = 376	“Keep your brain fit!” —online psychoeducational intervention.
Nebeker, C. and Zlatar, Z. (2021) [39]. USA	Exit Survey	To understand the motivations and perceptions of participants enrolled in studies promoting brain health and their opinions regarding the use of mHealth tools within this context.	65–80/*n* = 41	mHealth device to assist Independent Walking for Brain Health
Beristain et al. (2021) [40]. Consortium Spain, USA, Greece, Ireland, Italy	This paper describes the coaching aspects in CAPTAIN from different perspectives following a top-down approach.	To present a user-centered virtual coach for older adults at home to promote active and healthy aging and independent living.	Older adults/*	CAPTAIN—virtual coaching ecosystem (VCE).
Bilbao et al. (2016) [41]. Consortium UK, The Netherlands, Spain, Germany, Austria, France, Belgium	This paper describes the SONOPA framework and its elements, explains the deployments carried out in the course of the project execution and the evaluation of the matchmaking algorithm and draws some conclusions and proposes further work.	To present an Ambient Assisted Living framework developed within the SONOPA project, whose objective is to promote active aging by combining a social network with information inferred using in-home sensors.	Older adults/*	SONOPA—an Ambient Assisted Living framework.
Konstantinidis et al. (2014) [42]. Greece	This paper presents the design, implementation, wide deployment, and evaluation of the FitForAll platform system usability, user adherence to exercise, and efficacy are explored.	To present the design, implementation, wide deployment, and evaluation of the low cost, physical exercise, and gaming (exergaming) FitForAll (FFA) platform system usability.	67–87/*n* = 116	FitForAll (FFA) platform—maintain/advance healthy physical status and well-being.
Irvine et al. (2013) [43]. USA	A randomized controlled trial.	To evaluate the efficacy of a 12-week Internet intervention to help sedentary older adults adopt and maintain an exercise regimen.	Over 55/*n* = 368	Active After 55—participation in exercise activities on a regular basis.
Muellmann et al. (2019) [44]. Germany	A randomized controlled trial.	To examine the effects of two web-based interventions on physical activity in older adults compared to a delayed intervention control group.	65–75/*n* = 589	PROMOTE study—supported in web-based interventions, one including subjective physical activity monitoring and the other a combination of subjective and objective physical activity monitoring.
Ballesteros et al. (2014) [45]. Consortium Spain, Sweden, and Greece	A randomized controlled trial.	To investigate the potential of new Information and Communication Technology (ICT) environments to help maintain independence and well-being.	65–85/*n* = 57	AGNES is a novel ICT solution that promotes connectivity and social inclusion.
Compernolle et al. (2020) [46]. Belgium	A mixed methods study to evaluate engagement, acceptability, usability, and preliminary efficacy.	To evaluate engagement, acceptability, usability, and preliminary efficacy of a self-monitoring-based mHealth intervention developed to reduce sedentary behavior.	60/*n* = 28	Activator self-monitoring device of sedentary behavior.
O’Caoimh et al. (2018) [47]. Consortium The Netherlands and Italy	This paper describes the 25 healthcare related recommendations of PERSSILAA	To address pre-frailty and promote active and healthy aging, targeting three pre-frailty subdomains: nutrition, cognition, and physical function.	Older adults/*	PERSSILAA—ICT-supported platform to screen, assess, manage, and monitor community-dwelling older adults
Christophorou et al. (2016) [48]. Consortium Switzerland and The Netherlands	This paper described the Miraculous-Life project.	To identify and assess a set of services that an ICT system for Aging Well should support, in an actual end-user setting.	Older adults/1st trial–*n* = 7 older adults and *n* = 2 caregivers; 2nd trial–*n* = 16 older adults and *n* = 5 caregivers; 3rd trial–*n* = 15 older adults and *n* = 4 caregivers.	Miraculous-Life—a package of ICT services for Ageing Well that promotes ‘Positive Ageing’.
Taraldsen et al. (2020) [49]. Norway, Germany, and The Netherlands	A randomized controlled trial.	To assess the feasibility of delivering a lifestyle-integrated functional exercise program and evaluate the delivery of the intervention by use of digital technology (eLiFE) to prevent functional decline.	61–70/*n* = 180	eLiFE intervention—delivered to participants via the PreventIT application on a smartphone and a smartwatch through video clips, pictures, and text/verbal instructions for each activity.
Fornasini et al. (2020) [50]. Italy	Qualitative study to evaluate the usability of technologies and determine the user experience of participants.	To investigate the effectiveness of activities that combine geocaching and self-tracking technologies to promote active aging.	65–82/*n* = 14	The Impronte project—combines the geocaching activity, accompanied by gamification elements, with the use of fitness tracking technology in the form of a pedometer bracelet.

Legend: * lack of information about the data.

### 3.2. Description of the eHealth Platforms

Due to the demographic transformation caused by a rapidly aging population, ICT-based assistive technologies are needed to support older adults in staying active and independent, for as long as possible, in their chosen home environment.

Internet of Things technologies (wearable devices, environmental sensors) combined with analytics in the cloud provide a virtual coaching system to support healthy aging [37,40,41,45]. The NESTORE [37] system includes a plethora of sensors and other data sources (wearable devices, environmental sensors) to overcome the limitations of vertical solutions (single-domain oriented works) with a strong holistic approach by addressing several singular domains (cognitive, physiological, social, and nutritional) and by considering these domains from a user-centered approach. The CAPTAIN [40] virtual coaching ecosystem places older adults at the center of the coaching process.

The SONOPA [41] project aimed to promote active aging by combining social networks with information inferred using in-home sensors.

AGNES [45] is an ICT solution aimed at providing older people with technological support and social connectivity to keep them in touch with family and friends, avoiding isolation and promoting connectivity and social inclusion, through a home system equipped with sensors, specific web services, and dedicated web interfaces [45].

Online interventions are another mode of delivery to promote an active and healthy life [38,43,44,47,48]. The “Keep your brain fit!” online intervention was designed as a computer-generated personalized intervention [38].

The online intervention Active After 55 was designed to provide information and support to improve knowledge, attitudes, self-efficacy, and behavioral intentions to participate in exercise activities on a regular basis [43]. The PROMOTE project consists of two web-based interventions to promote physical activity among older adults. One group of participants received access to a web-based diary and were encouraged to track their behavior over a 10-week period. Another group of participants received a Fitbit Zip (Fitbit, San Francisco, CA, USA) to objectively track physical activities. Data of the Fitbit Zip were synchronized with the website at regular time intervals. The website provides weekly feedback on whether physical activity goals are reached and opportunities to network with other intervention participants via an invite-friends function and a forum [44]. PERSSILAA is a comprehensive ICT-supported platform to screen, assess, manage, and monitor community-dwelling older adults [47].

Christophorou et al. (2016) [48] identified and evaluate a set of services that an ICT system for Aging Well should support, in a real end-user configuration. This study was based on the Miraculous-Life project that designed, developed, and evaluated a Virtual Support Partner (by analogy to a real-life human partner). The system considers established behavior and communication patterns related to how an elder interacted with a human partner while carrying out his/her daily activities at home throughout life.

Other studies addressed the use of mHealth interventions based on self-monitoring to promote older adults’ active life [39,46,49].

The mHealth device-Independent Walking for Brain Health is a heart rate tracker that can be programmed through its app to set custom heart rate target zones equivalent to moderate intensity physical activity for each person. The device was programmed to vibrate and flash different colored lights (blue = below zone, green = in zone, red = zone above) when participants were deviating from their individually prescribed target heart rate zones [39].

The Activator is a mHealth intervention based on self-monitoring designed to reduce older adults’ sedentary behavior [46].

The eLiFE intervention was delivered to participants through the PreventIT app on a smartphone and a smartwatch via video clips, images, and text/verbal instructions for each activity [49].

Some platforms of gaming and exergaming have emerged as a response to the call for technology supporting active and healthy aging [42,50].

The FitForAll (FFA) platform offers specific exercises for older adults within an engaging game environment to promote adherence to the physical exercise protocol [42]. The Impronte project combined the geocaching activity, accompanied by some gamification elements, with the use of fitness tracking technology in the form of a pedometer bracelet [50].

### 3.3. Domains of Intervention of These eHealth Platforms

Aging imposes an economic burden on society due to the increasing needs resulting from age-related decline in physical, mental, and cognitive health. Healthy aging has become a public health priority to maintain the quality of life and autonomy of older adults [39,46].

Some authors report that regular physical activity among older adults delays functional decline, reduces the risk of chronic diseases and falls, improves quality of life, and helps them remain independent for as long as possible [43,44,50]. The promotion of healthy aging should involve several domains: physical, mental, emotional, and cognitive areas need to be considered when designing ICT-based solutions for older adults. However, most eHealth solutions available on the market are designed to address only a subset of these domains [37,38,40].

These results show that the available services seek to promote physical and cognitive activity, motivation for daily activities, socialization, and communication. While some platforms are multidimensional [37,38,39,40,47,48,50], others focus on a single domain of intervention [41,42,43,44,45,46,49] (Table 2). More detailed information follows in Appendix B.

### 3.4. Outcomes Assessed

Most of the studies (*n* = 6) focused on the age group of 65 and over. Although focusing on older adults, four studies did not specify an age limit. The remaining of the studies covered those who were 40 (*n* = 1), over 55 (*n* = 1) and over 60 years (*n* = 2). Most of the eHealth platforms focused on physical domains, specifically five out of 12 focused only on physical domains. Seven platforms focused on social domains, out of these only one focus solely on social aspects. Active ageing was mainly anchored on maintaining physical activity along with social interaction. Six of the platforms focused on the cognitive domain that included memory, and four on nutritional issues.

Some studies included in this scoping review described the process of designing eHealth platforms [37,39,40,46,47,48,50].

The included study on the NESTORE [37] system did not validate it. The authors suggested that its validation has to address the question of whether the proposed and developed system will indeed serve the intended purpose of having a positive impact on well-being and health. Moreover, it must show that the different components and tools are safe and practical in their use and offer information on the feasibility of carrying out a clinical trial to demonstrate the effectiveness of the system in promoting healthy aging, quality of life, and well-being. Concerning the CAPTAIN [40] ecosystem, the authors consider that the concept of virtual coaching presented in the included study will be part of future active and healthy aging systems for independent living, combined with other services, and become a milestone in the assisted living environment at home, which will extend the number of healthy years, reduce costs with health and social services, and increase the quality of life of older adults.

Other studies addressed both accessibility and usability [39,46,47,48,50]. The Activator [46] intervention was considered interesting, helpful, and easy to use, and was able to increase awareness among older adults of their sedentary behavior. The results of this study suggest that the innovative self-monitoring-based mHealth intervention holds potential for the reduction of sedentary behavior in older adults. In turn, Nebeker and Zlatar [39] argue that understanding the participants’ perceptions about the challenges and concerns introduced by mHealth is important, as acceptance will influence adoption and adherence to the study protocol. They believe that researchers should consider what might influence participants’ acceptance of the technology (access, data management, privacy, risks) and include them in the mHealth study design process. The PERSSILAA [47] project shows the potential to use an ICT-based multi-domain platform with pre-frail older adults. The results of this evaluation are being analyzed further and future research is being planned to validate the PERSSILAA platform with a suitably powered RCT to determine if ICT-supported services can prevent or delay the onset of frailty and functional decline in pre-frail community-dwelling older adults.

Furthermore, the services that encourage older adults’ socialization and communication with their family/friends/neighbors are very useful, as they increase older adults’ social interaction, prevent loneliness and isolation, increase their well-being, and improve their autonomy, independence, and confidence. The services that encourage older people to perform daily activities for a healthy and active life are also very important because they enable them to live actively and independently for longer in their home environment, thus delaying admission to hospitals and nursing homes [48].

The research developed by the Impronte project [50] adds that all users should actively participate in the creation of project contents from a peer-to-peer perspective. This approach can enable stakeholders, policy makers, and innovators to better understand the future of aging and address the challenges of innovation for older users more completely. Going beyond an accommodating vision that sees science and technology as mere solutions is critical for designing technologies that are functional for older adults (and their caregivers) and allowing them to embrace the challenge of active aging. Co-design with diverse seniors’ stakeholder groups is required to capture the elderly person’s perception of technology use. Such endeavor must attend to their perspectives on functional capacity, gender, professional and educational background, cultural differences, computer skills and experience [51].

The RCTs [38,43,44,45,49] correspond to studies conducted to assess the effectiveness of the intervention. After the online intervention “Keep your brain fit!” [38], the experimental group reported more feelings of stability in memory functioning and perceived greater locus of control over memory than the control group. Additionally, research conducted on the Active After 55 [43] intervention demonstrates that a stand-alone Internet-based exercise program that adapts content according to users’ preferences and interests can increase self-reported physical activity and be well-received by sedentary older adults. An RCT on the eLiFE platform [49] provided evidence that an ICT-based lifestyle-integrated exercise intervention, focusing on behavioral change, is feasible and safe for young older adults. These online interventions to promote physical activity seem to be cost-effective and can reach a large number of older people [38,43]. On the other hand, as described in the study protocol of Muellmann et al. [52], the aim of the study is to evaluate the effectiveness of two web-based interventions for the initiation and maintenance of regular PA (i.e., intervention groups 1 and 2) compared to a delayed intervention control group of older adults aged 65 to 75 years, being that, in the RCT [44], participation in the two web-based interventions did not lead to significant increases in moderate-to-vigorous physical activity or significant decreases in sedentary time compared to the group control. The RCT on the ICT-based project AGNES [45] found a statistically significant improvement in the user group from pre-test to post-test on the affective dimension of the well-being SPF-IL scale. This dimension is related to the degree of confidence, social acceptance, and level of satisfaction with the people around them.

Finally, other studies described the framework developed within the projects [41,42]. Bilbao et al. (2016) [41] described a new approach that promotes active aging with the use of sensor data and social networks, giving older adults the opportunity to create new social connections with similar people in order to curb social isolation and promote a healthy social life. Konstantinidis et al. (2016) [42] reported on the design, implementation, and thorough evaluation through a wide pilot deployment of the FFA exergaming platform for senior users. Formal assessment was carried out along two main axes, namely, on user and efficacy levels. On the usability front, this study provided strong evidence that such interventions are feasible and may be implemented through systems capable of motivating older adults to engage with a healthy physical activity program. On the efficacy axis, it provided strong evidence of physical improvement by means of clinical tests and improvement of the general wellness and quality of life of its user base.

## 4. Discussion

Caring for an aging population is one of the major challenges for the 21st century.

The increased interest and the acceptance of digital tools among health and social care providers, as well as consumers offers a unique and important opportunity to explore the effectiveness and full potential of various digital initiatives that assist the de-livery of support and health care [20,21].

Therefore, it is urgent to develop new technologies to help older people maintain their independence and improve their quality of life [6,16,53].

On the other hand, the benefits and barriers of eHealth are, in particular, crucial aspects concerning its implementation and adoption process. In this respect, Eysenbach (2001) [54] proposed a conceptual framework to frame the potential impact and the main factors of eHealth. In his seminal work, Eysenbach (2001) [54] listed and described the 10 essential e’s in eHealth: Efficiency, Enhancing quality, Evidence-Based, Empowerment, Encouragement, Education, Enabling, Extending, Ethics, and Equity. According to the author, the ‘e’ in eHealth does not mean just ‘electronic’ but implies a number of other ‘e’s’ that, together, best characterize what eHealth is or what it should be [54].

This scoping review identified several digital services: Internet of Things technologies (wearable devices, environmental sensors) [37,40,41,45]; online interventions [29,34,35,38,39]; mHealth interventions [39,46,49], and gaming and exergaming [42,50].

Today, ICT includes devices and applications that provide access to information and enable electronic communications, such as messaging and video chatting. Cell phones, smartphones, smartwatches, computers, and laptops are typical ICT devices. The Internet (e.g., the web) is another ICT and plays a special role, as it is not a stand-alone device but a network of numerous systems and devices [55,56]. Therefore, ICTs are critical for older people’s social inclusion, participation, and well-being, namely through the implementation of Healthy and Age-Friendly Environments (SHAFE) in community spaces where older people can remain active citizens [27,28,33].

ICT use in general may enable older individuals to live independently for a longer time and may have positive effects on health and social isolation [32,56]. The included studies [37,40,41,45] have shown that ICT enables new forms of social participation and interaction and improves access to information. On the other hand, some studies have also revealed several advantages of online interventions for physical activity, given that this type of intervention is potentially more cost-effective because it can reach a large number of people through the Internet [38,39,40,42,43,44,46,47,48,49,50]. The results of some studies [11,22] provide further support for recommending eHealth interventions as a viable strategy to reduce physical inactivity.

The outbreak of the COVID-19 pandemic has led to an acceleration in the development of web-based interventions to alleviate mental health-related impact [20,21]. According to the study by Shapira et al. [21] digital intervention programs should pay particular attention to older adults who live alone, as they may suffer more from social isolation and find it especially difficult to maintain social contacts during the pan-demic. This relatively simple model can be effectively utilized by communities globally to help connect lonely and isolated older inhabitants, both during the pandemic and more routinely.

Concerning the design and development of an eHealth platform for active and autonomous living, the specific needs and requirements of older adult users must be prioritized to ensure that accessibility and usability are considered (e.g., due to their hearing or vision impairment or limited ability to control digital devices) [6,57]. Moreover, the demographic differences and older people’s different perspectives must be taken into account. Bilbao et al. (2016) [41] and Ballesteros et al. (2014) [45] concluded that it is crucial to expand older people’s social circle through the use of sensor data and social networks, giving them the opportunity to develop new relationships to curb social isolation, promote a healthy social life and psychological well-being, and remain socially involved. Referring older people to social activities can help to avoid social isolation and improve their mental and physiological health (Buyl et al., 2020) [16].

From this perspective, Ienca et al. (2021) [58] found that effective digital health technologies for healthy aging can be interpreted as an empowering factor that facilitates aging in place. These patient-centered technologies, created to promote freedom and independent living, are more likely to be adopted and appreciated by community-dwelling older adults. A difficult barrier to overcome is the prior ignorance of the majority of older adults to the technology that would be overcome by working as a team throughout the community, especially in the health and educational sector, as well as the family or social nucleus [53,59].

According to Fornasini et al. (2020) and Beristain et al. (2021) [40], a user-centered virtual service and user involvement in the design and development of the platform are key to promoting active and healthy aging and independent living. Irvine et al. (2013) [43] also demonstrated that a theoretically-based stand-alone Internet exercise program that tailors content according to users’ preferences and interests can increase self-reported physical activity and be well received by sedentary older adults.

Although experimental designs such as RCTs to evaluate e-Health interventions are expensive and time-consuming, they are important to demonstrate their effectiveness and cost-effectiveness [16]. This scoping review identified six RCTs. Irvine et al. (2013) [43] found that this type of intervention can be made available to users on the Internet, making it a potentially cost-effective physical activity tool that can reach a large number of people. Reijnders et al. (2015) [38] argued that the relatively low costs and easy accessibility of this e-health intervention make it a valuable contribution to public healthcare interventions for middle-aged and older adults.

Beristain et al. (2021) [40] considered that the concept of virtual coaching in their paper will be part of the future active and healthy aging systems for independent living, combined with other services, and become a milestone in the assisted living environment at home, which will extend the number of healthy years, reduce costs with health and social services, and increase the quality of life of older adults.

Based on the results of this scoping review, it can be concluded that the majority of studies addressed the physical, social, and cognitive domains. In fact, except for the study by Reijnders et al. (2015) [38], all included studies considered at least one of these domains as inherent to the functioning of each eHealth platform. Some studies took into account these three domains in the platform under analysis [37,40,47]. These results are thus in line with the assumptions of the definition of Active Aging, a concept that involves the promotion of physical, social, and mental well-being throughout the life course and the participation in society, allowing individuals to successfully overcome the challenges [6,9,26].

Although cost-effectiveness analyses are not widely available, ICT-based interventions can be more cost-effective than traditional interventions [11,18,55]. However, studies indicate that these digital solutions can overcome barriers to access to healthcare, as they have the potential to reach a larger proportion of the population [40,43].

Another major factor identified as a point of failure in the development and implementation of health information systems is the limited understanding of users, their needs, and the contexts in which the systems are used. Therefore, it is important to understand end-user needs from multiple perspectives [32,58]. Ballesteros et al. (2014) [45] also stated that the degree of confidence, social acceptance, and level of satisfaction are critical to the successful adoption and diffusion of new technologies. In the study by Compernolle et al. (2020) [46], the self-monitoring-based mHealth intervention was considered interesting, useful, and easy to use, being able to increase the awareness of the elderly about their sedentary behavior. Posteriorly, an intervention study was carried out, with the aim of examining if older adults break their sedentary behavior immediately after receiving personalized haptic feedback on prolonged sedentary behavior and if the percentage of breaks differs depending on the time of the day when the feedback is provided. The results of this study that demonstrated a majority of haptic vibrations, especially those in the morning, did not result in a break in the sedentary behavior of older adults. As such, simply bringing habitual sedentary behavior into conscious awareness seems to be insufficient to target sedentary behavior [60]. In turn, O’Caoimh et al. (2018) [47] found that health literacy and ICT literacy (eHealth) were useful in helping older people to access such services. The results were supported by the study cohort multiple randomized controlled trail [56]. This study provided evidence that a home exercise program is easy to use and has the potential to improve the quality of life and health status of pre-frail older adults living at home [56]. Taraldsen et al. (2020) [49] assessed the feasibility, usability, and acceptability of eLiFE using an ICT platform, combining technology with behavior change techniques, and concluded that it was as successful as delivering the intervention using traditional paper manuals. On the other hand, Nebeker and Zlatar (2021) [39] found that all participants reported that the device helped them to walk faster, with 65% stating that it was easy to synchronize data from the device to the phone.

Thus, a user-centered design approach is recommended to fully address users’ health and technology needs through the development and implementation of these platforms.

In addition, the healthy aging perspective includes the physical, social, and cognitive domains, as well as healthy environments to promote healthy lifestyles, well-being, and social participation [25,60].

## 5. Conclusions

This scoping review aimed to map and understand the extent and type of evidence on eHealth platforms to promote active aging. It synthesized information on the type of eHealth platforms, the targeted age groups, the domains of intervention, and the outcomes assessed in the studies that have implemented and evaluated these eHealth platforms.

This information can be used to guide future research lines and serve as a basis for a systematic review of effectiveness, aiming to identify the most effective eHealth platforms to promote Active Aging.

Digital health designers and researchers should make every effort to ensure the involvement of older adults in the design process and research of digital technology. Based on this paper’s results, we suggest that ageism in the design process of digital technology might play a role as a possible barrier for adopting technology.

This scoping review may contribute to informing the different stakeholders about these eHealth platforms, whether they be patients, caregivers, health professionals, or policy makers.

## Figures and Tables

**Figure 1 ijerph-19-15940-f001:**
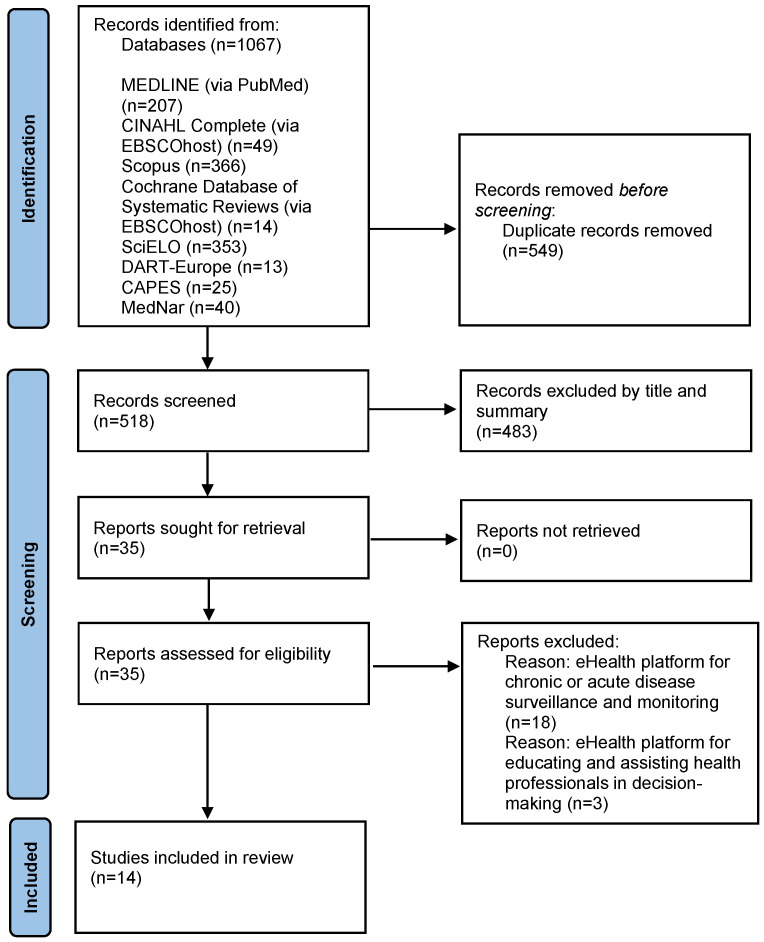
PRISMA flow diagram of the systematic review process [34].

**Table 2 ijerph-19-15940-t002:** Domain(s) of intervention.

Domain(s) of Intervention of eHealth Platforms	eHealth Platforms/Project
Physical	NESTORE [37]
	Independent Walking for Brain Health [39]
	CAPTAIN [40]
	FitForAll (FFA) [42]
	Active After 55 [43]
	PROMOTE [44]
	Activator [46]
	PERSSILAA [47]
	Miraculous Projet [48]
	eLiFE [49]
	Impronte Project [50]
Nutrition	NESTORE [37]
	CAPTAIN [40]
	PERSSILAA [47]
Cognition	NESTORE [37]
	Keep your brain fit [38]
	Independent Walking for Brain Health [39]
	CAPTAIN [40]
	PERSSILAA [47]
Social	NESTORE [37]
	CAPTAIN [40]
	SONOPA [41]
	AGNES [45]
	PERSSILAA [47]
	Miraculous Project [48]
	Impronte Project [50]

## Data Availability

For data supporting reported results please contact the authors of this review.

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
