# Peer review of "eHealth Platforms to Promote Autonomous Life and Active Aging: A Scoping Review"

_ijerph, 2022, doi:10.3390/ijerph192315940_

Round 1
Reviewer 1 Report
Thank you for inbiting me to review the current paper. I read it with interest and found it limited in the scope of papers included and conclusions. I hope my remarks will I, prove the paper.
“It is widely recognized that health systems must shift from reactive disease 68 management to health promotion and disease prevention, from a focus on illness to a 69 focus on well-being, and from fragmentation to the integration of services along the 70 continuum of care. These issues are not recognized in the European Union when only 3% 71 of the health budgets focus on promoting health, independence, and autonomy [16].
Expand on widely recognize and the shift of theory and conceptualization behind this recognition. This is the backbone of your review.
“Moreover, patient engagement is a broader concept that encompasses adherence, 73 empowerment, health literacy, shared decision-making, and patient activation.”
Expand
It seems that 83-98 needs to be moved to the head of the introduction.
Currently the introduction is lacking a logical and smooth development. It reads as bits of information are provided. Rethink your theory and reorganize the introduction.
You need to add studies conducted during COVID pandemic when elderly people undergo changes in their social life. Especially in the first waves before vaccinations were provided. related to the importance of having an ehealth accessible method when disasters in general occurred.
I suggest to read the following studies relevant to your paper: address the psychological intervention via digital platforms.
Expand on the method of using ehealth – individually and in a group setting:
Shapira, S., Yeshua-Katz, D., Goren, G., Aharonson-Daniel, L., Clarfield, A. M., & Sarid, O. (2021). Evaluation of a Short-Term Digital Group Intervention to Relieve Mental Distress and Promote Well-Being Among Community-Dwelling Older Individuals During the COVID-19 Outbreak: A Study Protocol. Frontiers in Public Health, 9, 577079.
Shapira, S., Cohn-Schwartz, E., Yeshua-Katz, D., Aharonson-Daniel, L., Clarfield, A. M., & Sarid, O. (2021). Teaching and Practicing Cognitive-Behavioral and Mindfulness Skills in a Web-Based Platform among Older Adults through the COVID-19 Pandemic: A Pilot Randomized Controlled Trial. International Journal of Environmental Research and Public Health, 18(20), 10563.
And see possible relevant theory
Mannheim, I., Schwartz, E., Xi, W., Buttigieg, S. C., McDonnell-Naughton, M., Wouters, E. J., & Van Zaalen, Y. (2019). Inclusion of older adults in the research and design of digital technology. International Journal of Environmental Research and Public Health, 16(19), 3718.
You need to look at the possible application of e health for mental health., Discuss possible differences from physical health.
Author Response
Dear Editor, we thank you for speeding up this process.
We appreciate the contribution made by this review process.
The detailed response to each of the reviewers follows.
Authors' response to Reviewer 1:
First of all, we thank you for the careful evaluation carried out, which has been an excellent contribution to improving the quality of the article. Regarding comments, we respond in the attached document with the description of the changes made, which are marked in the manuscript.

Reviewer 2 Report
Please see the attached file.

Author Response
Dear Editor, we thank you for speeding up this process.
We appreciate the contribution made by this review process.
The detailed response to each of the reviewers follows.
Authors' response to Reviewer 2:
First of all, we thank you for the careful evaluation carried out, which has been an excellent contribution to improving the quality of the article. Regarding comments, we respond in the attached document with the description of the changes made, which are marked in the manuscript.

Round 2
Reviewer 1 Report
Nicely revised I recommend to accept the manuscript
Reviewer 2 Report
The revised version is improved. I appreciate the authors' efforts to address my concerns.